# Autophagy Deficiency in Renal Proximal Tubular Cells Leads to an Increase in Cellular Injury and Apoptosis under Normal Fed Conditions

**DOI:** 10.3390/ijms21010155

**Published:** 2019-12-25

**Authors:** Chigure Suzuki, Isei Tanida, Juan Alejandro Oliva Trejo, Soichiro Kakuta, Yasuo Uchiyama

**Affiliations:** 1Department of Cellular and Molecular Neuropathology, Juntendo University Graduate School of Medicine, Bunkyo-Ku, Tokyo 113-8421, Japan; cgsuzuki@juntendo.ac.jp (C.S.); oliva@juntendo.ac.jp (J.A.O.T.); skakuta@juntendo.ac.jp (S.K.); 2Department of Cellular and Molecular Pharmacology, Juntendo University Graduate School of Medicine, Bunkyo-Ku, Tokyo 113-8421, Japan; 3Laboratory of Morphology and Image Analysis, Biomedical Research Center, Juntendo University Graduate School of Medicine, Bunkyo-Ku, Tokyo 113-8421, Japan

**Keywords:** autophagy, *Atg7*, renal proximal tubular cell, gene knockout mouse

## Abstract

Renal proximal tubular epithelial cells are significantly damaged during acute kidney injury. Renal proximal tubular cell-specific autophagy-deficient mice show increased sensitivity against renal injury, while showing few pathological defects under normal fed conditions. Considering that autophagy protects the proximal tubular cells from acute renal injury, it is reasonable to assume that autophagy contributes to the maintenance of renal tubular cells under normal fed conditions. To clarify this possibility, we generated a knock out mouse model which lacks Atg7, a key autophagosome forming enzyme, in renal proximal tubular cells (*Atg7^flox/flox^;KAP-Cre^+^*). Analysis of renal tissue from two months old *Atg7^flox/flox^;KAP-Cre^+^* mouse revealed an accumulation of LC3, binding protein p62/sequestosome 1 (a selective substrate for autophagy), and more interestingly, Kim-1, a biomarker for early kidney injury, in the renal proximal tubular cells under normal fed conditions. TUNEL (TdT-mediated dUTP Nick End Labeling)-positive cells were also detected in the autophagy-deficient renal tubular cells. Analysis of renal tissue from *Atg7^flox/flox^;KAP-Cre^+^* mice at different age points showed that tubular cells positive for p62 and Kim-1 continually increase in number in an age-dependent manner. Ultrastructural analysis of tubular cells from *Atg7^flox/flox^;KAP-Cre^+^* revealed the presence of intracellular inclusions and abnormal structures. These results indicated that autophagy-deficiency in the renal proximal epithelial tubular cells leads to an increase in injured cells in the kidney even under normal fed conditions.

## 1. Introduction

Renal proximal tubular epithelial cells reabsorb proteins from the glomerular filtrate that undergoes proteolysis by the lysosomal system [1,2]. Proximal tubular cells are susceptible to many types of injury including ischemia-reperfusion injury and nephrotoxic drugs/substances. Recently, we reported that renal ischemia/reperfusion injury results in an increase in kidney injury molecule (Kim-1), a marker for early kidney injury, in cathepsin D-deficient renal tubular epithelial cells [3]. Kim-1 is a type I transmembrane glycoprotein, and its expression is low in normal kidneys. Kim-1 is significantly increased in proximal tubule cells following kidney injury [4]. Upon renal injury, the extracellular domains of Kim-1 separate from the cell surface and enter the urine through a metalloproteinase-dependent process. Urinary Kim-1 concentration is markedly increased within 12 h following kidney injury [5]. Urinary Kim-1 is used as a clinical biomarker for the diagnosis of early acute kidney injury.

The autophagy-lysosome system is a bulk intracellular degradation system for organelles and lipids as well as proteins. When autophagy is induced, the isolation membrane appears, and extends to engulf intracellular components including organelles. Finally, the isolation membrane closes to form autophagosomes. The autophagosomes fuse with lysosomes to form autolysosomes. Autolysosomal contents are degraded by lysosomal hydrolases. Autophagy related (*Atg*) genes encode core components of the autophagy molecular machinery [6,7,8]. *Atg7*, one of the *Atg* genes, encodes a key E1-like enzyme essential for LC3 (microtubule associated protein 1 light chain 3)-lipidation and for Atg12-conjugation, both processes which are indispensable for autophagosome formation. When autophagy is activated, the cytosolic form of LC3 (LC3-I) is activated by Atg7, an E1-like enzyme [9,10,11], transferred to Atg3, an E2-like enzyme [12,13,14], and conjugated to phosphatidylethanolamine to form a membrane-bound form of LC3 (LC3-II, LC3-phosphatidylethanolamine conjugate) [15,16]. LC3-II is localized to autophagosomes.

Autophagy protects the cellular function of renal proximal tubular cells against injury induced by renal ischemia-reperfusion, nephrotoxic drugs, and ureteral obstruction injury. Nevertheless, there is no apparent defect in kidney function of renal tubular cell-specific autophagy-deficient mouse under normal fed conditions [17,18,19]. At present, there are few reports of intracellular abnormalities in autophagy-deficient renal proximal tubular cells under normal fed conditions. However, considering the increased sensitivity of autophagy-deficient renal tubular cells against acute kidney injury, it is possible that autophagy plays an indispensable role in maintaining normal functioning. To clarify the intracellular function of autophagy in the renal proximal tubular cells, we targeted the *Atg7* gene, and generated a renal proximal tubular cell-specific knockout mouse (*Atg7^flox/flox^;KAP-Cre^+^*).

## 2. Results

### 2.1. Newly Generated Atg7^flox/flox^;KAP-Cre^+^ Mice Exhibit Molecular Changes in Proximal Tubular Cells Including an Age-Dependent Increase in p62/SQSTM1 under Normal Fed Conditions

To analyze the effect of autophagy deficiency in renal proximal tubular cells, we generated proximal tubule-specific *Atg7* knockout (*Atg7^flox/flox^;KAP-Cre^+^*) mice by crossing *Atg7^flox/flox^* mice with kidney androgen-regulated protein (*KAP*)-*Cre*^+^ mice, and analyzed male mice. Since the *KAP* gene is expressed specifically in renal proximal tubule cells of the mouse fetal kidney during late pregnancy, *Cre* recombinase under the control of *KAP*-promoter is expressed in the renal proximal tubule cells during the development of kidney before birth [20]. In *Atg7^flox/flox^* mice, exon 14 of the *Atg7* gene is flanked by two *loxP* sequences [21]. Therefore, the *Atg7* gene of *Atg7^flox/flox^;KAP-Cre^+^* male mice is deleted specifically in renal proximal tubular cells. *Atg7^flox/flox^;KAP-Cre^+^* mice grew normal and fertile. Urinalysis, and blood biochemistry test results showed no significant difference between *Atg7^flox/flox^* and *Atg7^flox/flox^;KAP-Cre^+^* mice in their renal function consistent with previous report (Appendix A) [22]. *Atg7^flox/flox^;KAP-Cre^+^* seemed to have a lower urinary glucose level than control mice of the same age. Meanwhile, in general, the dysfunction of proximal tubular cells causes an increment of the urinary glucose level.

To confirm the deletion of *Atg7* in the kidney of *Atg7^flox/flox^;KAP-Cre^+^* mice, we prepared whole kidney lysates from two-month old mice and analyzed them with Western blotting. The results show that Atg7 was significantly decreased in the kidneys of *Atg7^flox/flox^;KAP-Cre^+^* mice compared with the kidneys of control mice (about 63% reduction) (Figure 1A). Since Atg7 is an E1-like enzyme essential for the LC3-lipidation, we also analyzed LC3 using Western blotting. The results show that lack of Atg7 results in accumulation of LC3-I (unconjugated LC3) in the kidney.

When autophagy is impaired, p62/SQSTM1 accumulates in the cells [23]. It has previously been reported that p62 accumulates in the proximal tubular cells of a nine-month old renal tubular cell-specific autophagy-deficient mouse [24]. To confirm if the tubular cells of our mouse model also exhibit the same feature, immunohistochemical staining was performed for p62 and megalin, a marker of renal proximal tubular cells, to examine renal tissue. Our results showed that p62-positive signals were observed in megalin-positive renal cells of two-month old *Atg7^flox/flox^;KAP-Cre*^+^ mice (Figure 1B,C). As shown in Appendix A, p62 aggregates did not merge with either LC3 or Lysosomal associated membrane protein 1 (Lamp1), a marker of degradative autophagy-lysosomal organelles, in the proximal tubular cells of *Atg7^flox/flox^;KAP-Cre^+^* mice. These results support the evidence that autophagy flux is impaired in proximal tubular cells of *Atg7^flox/flox^;KAP-Cre^+^* mice. To further investigate, we examined renal tissue from several age groups of *Atg7^flox/flox^;KAP-Cre^+^* mice. Our results showed that the number of p62-positive cells increased continuously with advancing age (Figure 1D).

### 2.2. Increases in Positive Immunosignals of Kim-1 and TUNEL in Normal Fed Atg7^flox/flox^;KAP-Cre^+^ Mice

Our previous results showed that p62 accumulated in the renal tubular cells of two-month old *Atg7^flox/flox^;KAP-Cre^+^* mice. Therefore, we assumed that there will be some intracellular defects in the mice under normal fed conditions. To investigate renal proximal tubular cell injury in *Atg7^flox/flox^;KAP-Cre^+^* mice, we employed Kim-1, a marker of early kidney injury. Immunohistochemical analysis revealed that Kim-1-positive signals localized to the lumenal surface of renal proximal tubular cells in *Atg7^flox/flox^;KAP-Cre^+^* mice. In contrast, the kidneys of control mice only showed a few positive signals (Figure 2A,B). Interestingly, in *Atg7^flox/flox^;KAP-Cre^+^* mice, Kim-1-positive signals were detected in p62-positive renal tubular cells, suggesting a possible relationship between p62 accumulation and tubular cell injury. (Figure 2C). Also, as expected, Kim-1 and p62 positive signals were almost absent in female *Atg7^flox/flox^;KAP-Cre^+^* mouse, (Appendix A). Additional evaluation of renal tissue from mice from different age groups revealed that Kim-1-positive signals increased continuously as the mice age (Figure 2D). This observation corresponded to our previous result where p62 accumulated in an age dependent manner. Together, these results indicated that under normal fed conditions, autophagy-deficiency in renal tubular cells led to an increase in the number of damaged cells in an age dependent manner.

Kim-1 increases in the renal proximal tubular cells when the kidney is damaged during acute kidney injury. Tubular cell apoptosis is also known to occur during acute kidney injury [25]. Thus, we hypothesized that because there was evidence of kidney injury, we would also find apoptotic tubular cells. Therefore, we investigated whether or not, renal tubular cell apoptosis was induced in *Atg7^flox/flox^;KAP-Cre^+^* under normal fed conditions using Terminal transferase-mediated dUTP nick end labeling (TUNEL) staining. Our results revealed many TUNEL-positive renal tubular cells in cortico-medullary area in two-month old *Atg7^flox/flox^;KAP-Cre^+^* mouse kidney (Figure 3). These results suggested that even under normal fed conditions, autophagy-deficiency in the renal proximal tubular cells resulted in cell injury and apoptosis.

### 2.3. Accumulation of Abnormal Structures in the Renal Proximal Epithelial Tubular Cells of Atg7^flox/flox^;KAP-Cre^+^ Mice

So far, we found that injured and apoptotic cells increased in number in autophagy-deficient renal tubular cells. In brain and liver tissues, autophagy-deficiency leads to an accumulation of abnormal autophagic structures [21,27], suggesting the possibility that abnormal intracellular structures accumulate in autophagy-deficient renal tubular cells. Toluidine blue staining of the renal tubules showed that some renal tubular cells in *Atg7^flox/flox^;KAP-Cre^+^* kidneys contained aggregates that dyed deeply with toluidine blue, whereas these aggregates were rarely observed in *Atg7^flox/flox^* renal tubular cells (Figure 4). Ultrastructural analysis of renal proximal tubular cells showed the accumulation of large amorphous structures that corresponded to toluidine blue-positive aggregates (Figure 5A,B). These large amorphous structures were composed of a large number of degenerated mitochondria (Figure 5B; asterisk), and vacuoles and tubulo-vesiclar structures. In addition, lamellarly/concentrically arranged membrane structures were also detected in the renal tubular cells (Figure 5D). These lamellarly-arranged membrane structures contained a high number of peroxisome-like structures (Figure 5E; asterisk) and deformed lysosome-like structures (Figure 5F; arrow).

## 3. Discussion

Our study is the first to show that under normal fed conditions, autophagy-deficiency in the renal proximal tubular cells resulted in an increase in the number of injured and apoptotic cells. Notably, the number of injured cells continually increased in an age-dependent manner. Moreover, autophagy-deficient renal proximal tubular cells exhibited accumulation of abnormal aggregates. All of these pathological changes were observed even without inducing additional renal injury.

Previous studies of autophagy deficiency in proximal tubular cells have shown how these cells are more sensible to injury induced by experimental renal disease models. [24,28,29,30]. These disease models are examined using tubular cell-specific autophagy-deficient mice at two months old (or more). Using *Atg7^flox/flox^;KAP-Cre^+^* mice, we showed that autophagy-deficiency in the proximal tubular cells led to increase in number of injured and/or apoptotic cells under normal fed conditions, even in the absence of outstanding blood and urinary test results. These increases in the number of damaged cells may help explain why previous studies have reported that autophagy deficient proximal tubular cells exhibit higher sensitivity to injury. In the kidney, glucose filtered through the glomerulus is retrieved by proximal tubular cells. Therefore proximal tubular cell dysfunction is often accompanied by an increase in urinary glucose. Conversely, the urinary glucose level of *Atg7^flox/flox^;KAP-Cre^+^* mice was lower than same age *Atg7^flox/flox^* mice. At least, the reabsorption of glucose in the kidney of *Atg7^flox/flox^;KAP-Cre^+^* mice may not be impaired. At present, there is no answer for the results. To clarify this point, further studies will be required.

Kim-1 immunoreactivity is significantly increased in proximal tubule cells following kidney injury [4]. Female *Atg7^flox/flox^;KAP-Cre^+^* mice have no Kim-1 increment in their proximal tubular cells, providing further evidence that Atg7 deletion in male *Atg7^flox/flox^;KAP-Cre^+^* mice is the cause of proximal tubule cells injury. We found that autophagy-deficiency resulted in an increase of Kim-1-positive signals in the proximal tubular cells. Urinary Kim-1 concentration is markedly increased within 12 h following kidney injury [5]. The level of urinary Kim-1 is regarded as a clinical biomarker for the diagnosis of early acute kidney injury. It is therefore reasonable to assume that the increase in Kim-1 positive signals in the autophagy-deficient renal tubular cells of *Atg7^flox/flox^;KAP-Cre^+^* mice reflect an increase in the number of the injured cells.

There is a question from our results that needs addressing. If we could detect injured and apoptotic renal proximal tubular cells in *Atg7^flox/flox^;KAP-Cre^+^* mice, why those pathological changes did not reflect in the urinalysis and blood biochemistry test results? In general, when the renal proximal tubular cells are injured, severely injured cells are detached from the epithelial layer and fall into the luminal space of proximal tubules [31,32,33]. Simultaneously, new proximal tubular cells regenerate and compensate the function of the detached cells. These results indicate that this compensation may be sufficient to maintain kidney function under normal fed conditions.

Collectively, our results showed that renal proximal tubular cells were injured by inhibition of autophagy as evidenced by intracellular changes and cell death. However, these changes occur even in the absence of detectable defects in urinary and blood examinations. Moreover, the proximal renal tubular alterations following autophagy inhibition were also exacerbated with age. These events suggest that the use of drugs that suppress autophagy may impact renal function, and may be especially concerning in cases of long-term treatment.

## 4. Materials and Methods

### 4.1. Animal Model

All animal experiments were performed in accordance with guidelines of the Laboratory Animal Experimentation of Juntendo University (project license no. 290197, 1 April 2017) and approved by the Institutional Animal Care and Use Committee of Juntendo University. *Atg7^flox/flox^* mice and *KAP-Cre^+^* mice were kindly provided by Doctors Masaaki Komatsu (University of Niigata, Niigata, Japan) and Taiji Matsusaka (University of Tokai, Kanagawa, Japan), respectively. The renal proximal tubular cell-specific *Atg7*-KO mouse was established by breeding *Atg7^flox/flox^* mice [21] with *KAP-Cre^+^* transgenic mice [34]. Briefly *Atg7^flox/flox^* mice were bred to *KAP-Cre^+^* mice to obtain *Atg7^flox/+^*; *KAP-Cre^+^* mice and then *Atg7^flox/+^*; *KAP-Cre^+^* mice were crossed with *Atg7^flox/flox^* mice to get *Atg7^flox/flox^;KAP-Cre^+^* and *Atg7^flox/flox^* litter mates served as control. *KAP*-*Cre^+^* carries a promoter segment of *KAP* and cDNA encoding a *Cre* recombinase. Efficiency of *Cre* recombinase-mediated recombination in the *KAP-Cre^+^* mice was about 80% of proximal tubular cells of the S3 segment [34]. Since the *KAP* promoter is androgen dependent, only male mice were used in this study. KAP is supposed to be initially detected three days before birth [20].

### 4.2. Urinary Examinations

Urine samples were collected over a 24-h period using metabolic cages (3600M21, TECNIPLAST, Tokyo, Japan). Urinary total protein, albumin, potassium, sodium, glucose, urinary creatinine were analyzed by Oriental Yeast CO.LTD laboratory (Shiga, Japan).

### 4.3. Blood Examinations

Blood was collected from aorta and centrifuged for 15 min at 3000 rpm. Then the supernatant was retained as serum. Serum total protein, urea nitrogen, creatinine, uric acid, sodium, and potassium were analyzed by the Oriental Yeast CO.LTD laboratory.

### 4.4. Antibodies

Rabbit anti-LC3 and anti-Atg7 antibodies were purchased from Cell Signaling Technology (Danvers, MA, USA, cat#12741 and #8540), guinea pig anti-p62 antibody was form PROGEN Biotech GmbH (Heidelberg, Germany, cat#GP62-C), mouse anti-Megalin antibody was from NOVUS (Littleton, CO, USA, cat#NB110-96417), goat anti-Kim-1 antibody was from R&D systems (Minneapolis, MN, USA, cat#AF1817), and rabbit anti-Lamp1 was from Abcam (Tokyo, Japan, cat# ab24170). Donkey Alexa 488- conjugated anti-goat IgG, anti-mouse IgG, anti-rabbit IgG, and Donkey Cy3- conjugated anti-guinea pig IgG antibodies (Jackson Immuno Research, Philadelphia, PA, USA, cat#705-545-003 and 706-165-148) 4′,6-Diamidino-2-Phenylindole (DAPI) was purchased from Thermo Fisher Scientific (Waltham, MA, USA, cat#D1306).

### 4.5. Fixation and Embedding for Light and Electron Microscopy

For light microscopy, after being anesthetized, mice were fixed by cardiac perfusion with 4% paraformaldehyde buffered with 0.1 M phosphate buffer (PB), pH 7.2 [27,35]. After perfusion, renal tissues were excised and cut transversely and then immersed into the same fixatives for a further 24 h. These tissues were embedded in paraffin and cut at 5 μm. The sections were deparaffinized and stained with hematoxylin and eosin.

For electron microscopy, intracardiac perfusion with 2% paraformaldehyde and 2% glutaraldehyde buffered with 0.1 M PB, pH 7.2 was used to fix kidneys [27,35]. Fixed renal tissues were cut into small pieces, dehydrated with a graded series of alcohols and embedded in epoxy resin (TAAB Epon 812, EM Japan, Tokyo, Japan). Silver sections were cut with an ultramicrotome (Ultracut UCT, Leica, Nussloch, Germany), stained with uranyl acetate and lead citrate, and observed with an electron microscope (JEM-1230, JOEL, Tokyo, Japan). For toluidine blue staining, 0.5 μm semithin sections were used. In total, 5 random fields of cortico-medullary area were photographed, and deep-dyeing aggregates were counted. Data are expressed as the percentage of aggregates possessing tubular cells in total tubular cells.

### 4.6. Immunohistochemistry

As described previously [35], kidneys were fixed with 4% paraformaldehyde perfusion, excised, and frozen in OCT (Tissue-Tek, Torrence, CA, USA) compound to prepare 4 μm sections. Frozen sections were washed with 1× PBS and immunostained with anti-Kim-1, anti-KAP, anti-KDEL or anti-p62 antibodies. For quantification of Kim-1 and p62 positive areas, 5 random field images (20×) of cortico-medullary area were selected randomly. Kim-1- and p62-positive signals were quantified using Image J software 1.52a [26] (https://imagej.nih.gov/ij/index.html). These signals were calculated as the ratio of the positive signals per total area.

### 4.7. TUNEL Staining

For TUNEL staining we used paraformaldehyde-fixed paraffin-embedded kidney samples. After 5-μm thick paraffin sections were incubated with 20 µg/mL proteinase K (Roche) for 15 min at room temperature, TUNEL reaction were performed by using the In Situ Cell Death Detection Kit (Roche Diagnostics Corp., Rotkreuz, Switzerland) following the manufacturer’s instructions. Then DAPI staining was performed. For quantification, 5 random fields (20×) of cortico-medullary area were selected randomly and photographed using a confocal microscope (OLYMPUS FV1000, Tokyo, Japan) and TUNEL positive cells were counted.

### 4.8. Immunoblot Analysis

Kidneys were extracted from mice and independently homogenized with a Politron homogenizer in a lysis buffer consisting of 0.05 M Tris-HCl, pH 7.5, 0.15 M NaCl, and 1% Triton X-100 (Nacalai Tesque, Kyoto, Japan) with protease inhibitor cocktail (×100) (Nacalai Tesuque, cat#25955-11) for 30 min on ice. For quantification, Western blotting results were scanned and analyzed for signal intensity with Image J software.

### 4.9. Statistical Analysis

The obtained data from each experiment were expressed as the mean ± SEM. Statistical comparisons between the groups were performed using a student’s *t*-test, and statistical significance was set at *p* < 0.05. For Figure 4B, statistical analyses were performed using python software with scipy.stats module. Data in the graph are expressed as the median ± RANGE.

## Figures and Tables

**Figure 1 ijms-21-00155-f001:**
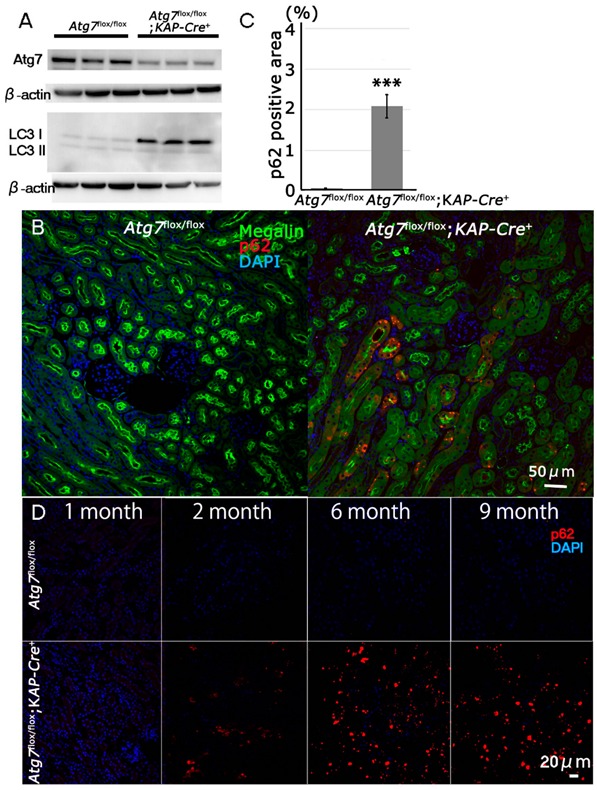
Impairment of autophagy in *Atg7*-deficient renal proximal tubular cells. (**A**) Decrease of Atg7 and increase of LC3-I in 2-month-old *Atg7^flox/flox^;KAP-Cre^+^* mouse kidney. Atg7 and LC3 Western blot (LC3-I/unconjugated LC3 and LC3-II/lipidated LC3) using whole kidney lysate of *Atg7^flox/flox^;KAP-Cre^+^* mice. As a control, *Atg7^flox/flox^* mouse kidney was employed. The intensity of each band of Atg7 and β-actin was estimated by densitometry. The ratio of Atg7 to β-actin was calculated. The amount of Atg7-positive signals in the *Atg7^flox/flox^;KAP-Cre^+^* mice kidney was about 63% lower than that in the *Atg7^flox/flox^* mice kidney (*n* = 3). Note that LC3-I was increased in the *Atg7^flox/flox^;KAP-Cre^+^* mouse kidney. (**B**) The massive accumulation of p62 in kidneys of 2-month old *Atg7^flox/flox^;KAP-Cre^+^* mouse. The cortico-medullary region of each kidney in 2 month-old *Atg7^flox/flox^;KAP-Cre^+^* mouse and *Atg7^flox/flox^* mouse was recognized with anti-p62 antibody (red). As a marker of renal proximal tubular cells, megalin (green) was employed. Nuclei were stained with 4′,6-diamidino-2-phenylindole (DAPI; blue). (**C**) Quantification of p62 positive area of 2-month-old *Atg7^flox/flox^;KAP-Cre^+^* (*n* = 5) and *Atg7^flox/flox^* kidney (*n* = 4, *** *p* < 0.01). Data in graphs are expressed as the mean ± SEM. Statistical analyses were performed using a student’s *t*-test. (**D**) Age-dependent accumulation of p62 (red) in the *Atg7^flox/flox^;KAP-Cre^+^* mouse kidney in 1-, 2-, 6-, and 9-month-old *Atg7^flox/flox^;KAP-Cre^+^* (lower) and *Atg7^flox/flox^* (upper) mouse kidney.

**Figure 2 ijms-21-00155-f002:**
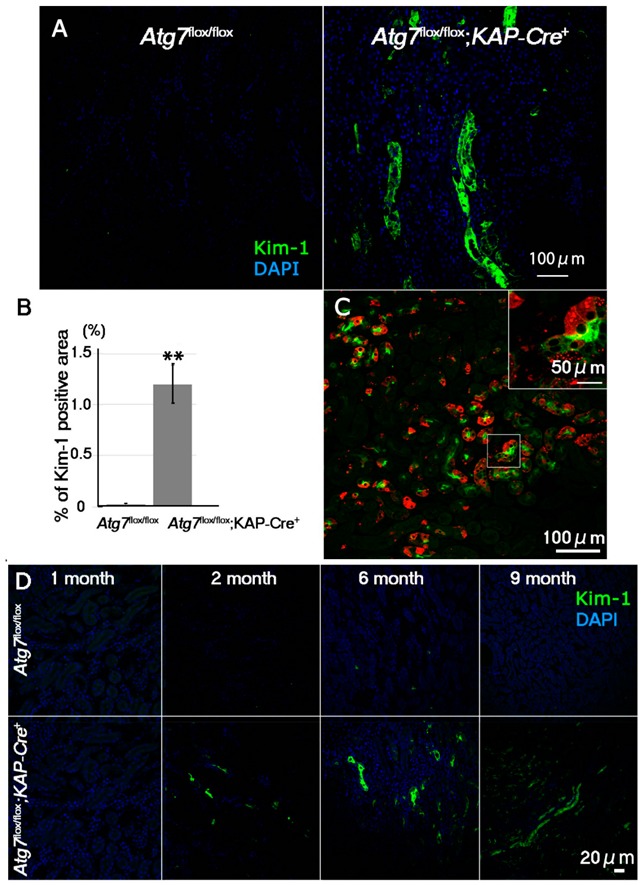
Age-dependent increase of Kim-1, a marker for early kidney injury, in the *Atg7^flox/flox^;KAP-Cre^+^* mouse kidney. (**A**) Representative images of Kim-1 immunostaining (green) in the proximal region of the kidneys of 2-month-old *Atg7^flox/flox^;KAP-Cre^+^* mice. As a control, *Atg7^flox/flox^* mouse kidneys were used. Nuclei were stained with DAPI (blue). (**B**) Quantification of Kim-1-positive area of 2-month-old *Atg7^flox/flox^* and *Atg7^flox/flox^;KAP-Cre^+^* mouse kidneys. Kim-1-positive signals in the cortico-medullary region of 2 month-old *Atg7^flox/flox^* (*n* = 4) and *Atg7^flox/flox^;KAP-Cre^+^* (*n* = 7) mice were estimated by ImageJ software (https://imagej.nih.gov/ij/index.html) [26]. Statistical analyses were performed using a student’s *t*-test (** *p* < 0.03). Data in the graph are expressed as the mean ± SEM. (**C**) Accumulation of Kim-1-positive signals in p62-positive autophagy-deficient renal tubular cells. Renal tubular cells in the cortico-medullary region were detected anti Kim-1 (green) and p62 (Red). (**D**) Age-dependent accumulation of Kim-1 in *Atg7^flox/flox^;KAP-Cre^+^* mouse kidney. Kidney samples from 1-, 2-, 6-, and 9-month-old *Atg7^flox/flox^;KAP-Cre^+^* (lower) and *Atg7^flox/flox^* (upper) mice were stained with anti-Kim-1 antibody (green).

**Figure 3 ijms-21-00155-f003:**
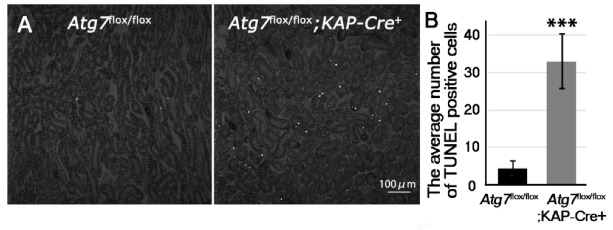
Increase of apoptosis in the cortico-medullary region of *Atg7^flox/flox^;KAP-Cre^+^* mouse kidney. (**A**) TUNEL staining in the cortico-medullary region of 2-month-old *Atg7^flox/flox^* and *Atg7^flox/flox^;KAP-Cre^+^* kidney. Apoptosis in the cortico-medullary region of *Atg7^flox/flox^;KAP-Cre^+^* mouse kidney was detected using TUNEL staining. (**B**) Quantification of TUNEL positive cells of 2-month-old mice. TUNEL-positive signals were evaluated with ImageJ software (*Atg7^flox/flox^*
*n* = 4 and *Atg7^flox/flox^; KAP-Cre^+^*
*n* = 5 *** *p* < 0.01). Statistical analyses were performed using a student’s t-test. Data in the graph are expressed as the mean ± SEM.

**Figure 4 ijms-21-00155-f004:**
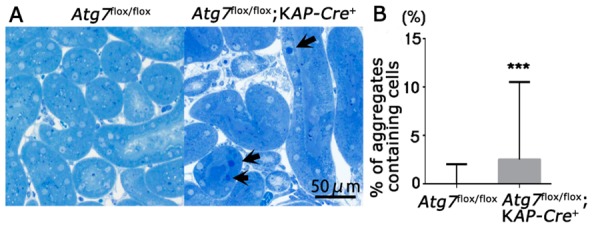
Accumulation of intracellular aggregates in the autophagy-deficient proximal tubular cells. (**A**) Representative images of renal proximal tubular cells in the cortico-medullary region in the kidneys from 2-month-old *Atg7^flox/flox^* and *Atg7^flox/flox^;KAP-Cre^+^* mice. The proximal tubular cells were stained with toluidine blue. Arrows indicate representative aggregates heavily dyed with toluidine blue. (**B**) Quantification of cells with toluidine blue positive aggregates dyed with toluidine blue. In each group at least five independent images were analyzed (magnification; 200×). Data in the graph are expressed as the median ± range. *** *p* < 0.01.

**Figure 5 ijms-21-00155-f005:**
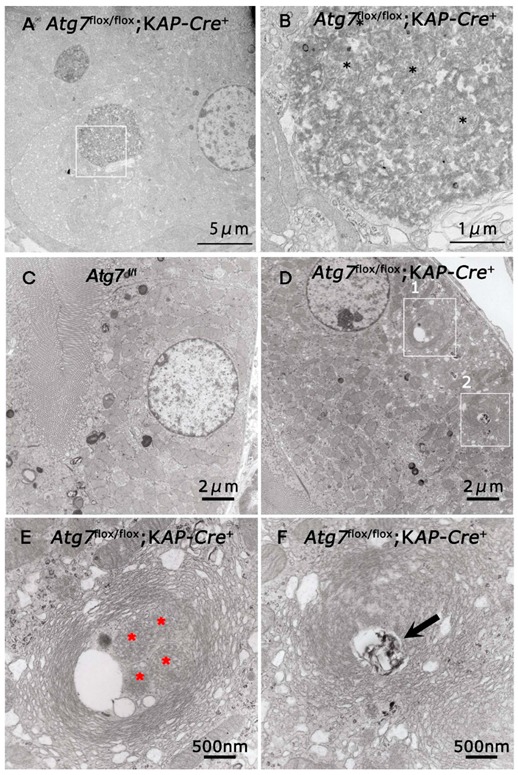
Electron microscopic analyses of *Atg7* deficient renal proximal tubular cells. The renal proximal tubular cells of 2-month-old *Atg7^flox/flox^;KAP-Cre^+^* (**A**,**B**,**D**–**F**) and *Atg7^flox/flox^* (**C**) mice were investigated with a transmission electron microscopy. (**B**) is higher magnification of area. Black asterisks indicated degenerated mitochondria. (**A**) (white box). (**E**,**F**) are higher magnification of areas (white boxes 1 and 2, respectively) in (**D**). Asterisks in (**E**) indicated peroxisome-like structures in the multi-lamellar bodies. Arrows indicate lysosome like structures in the multi-lamellar bodies.

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
