# Peer review of "Autophagy Deficiency in Renal Proximal Tubular Cells Leads to an Increase in Cellular Injury and Apoptosis under Normal Fed Conditions"

_ijms, 2019, doi:10.3390/ijms21010155_

Round 1

Reviewer 1 Report

In this report, Suzuki and colleagues present studies evaluating the phenotype of proximal-tubule deleted Atg7 mice.  Atg7 is a key autophagosome-forming enzyme.  They found that untreated KO mice had evidence of early renal injury including greater expression of KIM1.  This finding was associated with reduced autophagic markers (as expected).  The KO mice also had increased TUNEL positive staining suggesting greater degree of apoptosis with the absence of autophagy.  Electron microscopy allowed for finer analysis of aggregate structures remaining in the KO mouse kidney.  The paper is key in demonstrating the need for autophagy in the normal, healthy fed state to preserve proximal tubule structure and integrity and in the absence of this regulatory function, kidneys begin to develop signs of injury.  Additional markers of PT stress could be added to bolster the idea that the cells are not only accumulating protein aggregates, but also developing injury.  Some suggestions include measuring mitochondrial oxidative stress, collagen and glycogen deposition, and perhaps, transforming growth factor beta.  Additional comments below.

1. Abstract- please define p62/SQSTM1.

2. Supplementary data- blood glucose looks to be approaching significantly lower in the KO. Can you please provide the p-value within the figure?

3. Western blotting for KIM1 would add more depth to the finding, especially with regard to increases in expression with age, which were not quantified in the immunofluorescence.

4. The number of animals studied was very low (n = 3, Figure 2). Would be more confident with the conclusions if the n approached at least 6.  

5. Some comment regarding whether these findings are expected to translate to female mice is warranted in the Discussion.

Author Response

Response to the reviewer #1’s comments;

Thank you for your comments regarding our manuscript.

Major comments

Comment 1- Abstract- please define p62/SQSTM1.

Response: We added the definition of p62/SQSTM1 as ‘ubiquitin binding protein P62/sequestosome 1, one of the selective substrates for autophagy’ (Abstract, line 24)

Comment 2- Supplementary data- blood glucose looks to be approaching significantly lower in the KO. Can you please provide the p-value within the figure?

Response: Thank you for your advice. We replaced the graphs describing the p-value within the figure (supplementary data 1). We revised statistical analysis of blood and urinary data and represented them as median value in a bar graph (following Reviewer’s #2 comment #1). As you pointed out, 12 months old Atg7flox/flox;KAP-Cre+ showed significantly less urinary glucose than control (Atg7flox/flox) mice. In general, dysfunction of proximal tubular cells causes an increment of the urinary glucose level. So the results suggest that, at least, the reabsorption of glucose in the kidney of Atg7flox/flox;KAP-Cre+ mice may not be impaired.

We described about these observations in the results as ‘Atg7flox/flox;KAP-Cre+ seemed to have a lower urinary glucose level than control mice of the same age, while, in general, dysfunction of proximal tubular cells causes an increment of the urinary glucose level. ‘ (line 82-83 ) and discussion as ‘In the kidney, glucose filtered through the glomerulus is retrieved by proximal tubular cells. Therefore proximal tubular cell dysfunction is often accompanied by an increase in urinary glucose. Conversely, the urinary glucose level of Atg7flox/flox;KAP-Cre+ mice was lower than same age Atg7flox/flox mice. At least, the reabsorption of glucose in the kidney of Atg7flox/flox;KAP-Cre+ mice may not be impaired. At present, there is no answer for the results. To clarify this point, further studies will be required.’ (line 210-214) sections.

Comment 3-Western blotting for KIM1 would add more depth to the finding, especially with regard to increases in expression with age, which were not quantified in the immunofluorescence.

Response: We performed immunoblot analysis of p62 and Kim-1 in the kidney of between Atg7flox/flox and Atg7flox/flox;KAP-Cre+ mouse. However there was no significant difference between the two, since the damaged tubular cells of cortico-medullary region are limited to small parts of the kidney compared to whole renal tissue.

We further tried to isolate renal proximal tubular cells form the mouse kidney by using FACS. However, it was difficult to perform immunoblot analysis of p62 and Kim-1, since only a small amount of renal cells was isolated.

Comment 4-The number of animals studied was very low (n = 3, Figure 2). Would be more confident with the conclusions if the n approached at least 6.

Response: We performed additional experiments and added the data from one Atg7flox/flox and two Atg7flox/flox;KAP-Cre+ additional mice (Fig 1C, Fig 2B, and Fig 3B). Accordingly, the data from Figure 2B changed from three Atg7flox/flox to four mice, and Atg7flox/flox;KAP-Cre+ changed from five mice to seven mice.

Comment 5-Some comment regarding whether these findings are expected to translate to female mice is warranted in the Discussion.

 Response: We added immunostaining images of p62 and Kim-1 in female Atg7flox/flox;KAP-Cre+ mouse kidney (supplementary figure 3). And commented about this result in the Results as ‘Kim-1 and p62 positive signals were almost absent in female Atg7flox/flox;KAP-Cre+ mouse, (Supplementary Fig. S3). ‘(line 129-130) and Discussion sections as ‘Female Atg7flox/flox;KAP-Cre+ mice have no Kim-1 increment in their proximal tubular cells, providing further evidence that Atg7 deletion in male Atg7flox/flox;KAP-Cre+ mice is the cause of proximal tubule cells injury. ‘ (line 215-217).

Reviewer 2 Report

This is a very interesting manuscript, showing that genetic deletion of Atg7 per se leads to increase in injured cells in the kidneys.

The manuscript is clearly written, the experiments seem to be properly performed.

The results have been analysed presenting them as mean ± However, due to the low number of replicates (n = 3), it would be preferable to show the median value with its range or with the single sample value. Analysis of urine to determine renal function: Some parameters (glucose and volume in panel B) show a tendency to be different between the two types of mice. Here it would be important to increase the number of observations. I suggest to express urinary excretion as total excretion of substance under examination in 24 h urine/ body weight.

Minor: Fig. 1:

It would be desirable that also in the right part of panel B a glomerulus is visible.

 Is the exposure time for what shown in panel D the same? The DAPI labelling seems to be much more intensive in the lower part of panel D than in the upper part.

Author Response

Response to the reviewer #2’s comments;

We appreciate your comments and suggestions. Our responses to your comments are as follows:

Major comments

Comment 1-The results have been analyzed presenting them as mean ± However, due to the low number of replicates (n = 3), it would be preferable to show the median value with its range or with the single sample value.

 Response: We performed additional experiment and added data from one Atg7flox/flox and two Atg7flox/flox;KAP-Cre+ mice (Fig 1C,Fig 2B and Fig 3B). And as for blood and urinary data please see ‘comment 2’.

Comment 2-Analysis of urine to determine renal function: Some parameters (glucose and volume in panel B) show a tendency to be different between the two types of mice. Here it would be important to increase the number of observations. I suggest to express urinary excretion as total excretion of substance under examination in 24 h urine/ body weight.

 Response: Thank you for your advice. Unfortunately, we did not measure the body weight of the mice when we took urinary samples, so we cannot report the data as you suggested. Following your suggestion (comment 1), we revised the statistical analysis of blood and urinary data and represented them as a bar graph of median value ± range. As you pointed out, 12 months old Atg7flox/flox;KAP-Cre+ showed significantly less urinary glucose than control (Atg7flox/flox) mice. In general, dysfunction of proximal tubular cells causes an increment of the urinary glucose level. So the results suggest that, at least, the reabsorption of glucose in the kidney of Atg7flox/flox;KAP-Cre+ mice may not be impaired.

We described about these observations in the results as ‘Atg7flox/flox;KAP-Cre+ seemed to have a lower urinary glucose level than control mice of the same age, while, in general, dysfunction of proximal tubular cells causes an increment of the urinary glucose level. ‘ (line 82-83 ) and discussion as ‘In the kidney, glucose filtered through the glomerulus is retrieved by proximal tubular cells. Therefore proximal tubular cell dysfunction is often accompanied by an increase in urinary glucose. Conversely, the urinary glucose level of Atg7flox/flox;KAP-Cre+ mice was lower than same age Atg7flox/flox mice. At least, the reabsorption of glucose in the kidney of Atg7flox/flox;KAP-Cre+ mice may not be impaired. At present, there is no answer for the results. To clarify this point, further studies will be required.’ (line 210-214) sections.

We replaced the graphs with the p-value within the figure (supplementary data 1).

Minor comments

Comment-3: Fig. 1: It would be desirable that also in the right part of panel B a glomerulus is visible.

Response: We replaced it with a new image (Fig.1 B).

Comment-4: Is the exposure time for what shown in panel D the same? The DAPI labelling seems to be much more intensive in the lower part of panel D than in the upper part.

Response: We changed the image to a more suitable one (Fig.1 D).